# Efficacy of Clinical Guidelines in Identifying All Japanese Patients with Hereditary Breast and Ovarian Cancer

**DOI:** 10.3390/ijerph19106182

**Published:** 2022-05-19

**Authors:** Eri Haneda, Ann Sato, Nobuyasu Suganuma, Yoshiko Sebata, Saki Okamoto, Soji Toda, Kaori Kohagura, Yuka Matsubara, Yuko Sugawara, Takashi Yamanaka, Toshinari Yamashita, Satoru Shimizu, Hiroto Narimatsu

**Affiliations:** 1Department of Genetic Medicine, Kanagawa Cancer Center, Yokohama 241-8515, Japan; e-haneda@kcch.jp (E.H.); a_satou@kcch.jp (A.S.); n-suga@vesta.dti.ne.jp (N.S.); yoshiko.s.fp@kxe.biglobe.ne.jp (Y.S.); shimizustr0110@gmail.com (S.S.); 2Department of Breast and Endocrine Surgery, Kanagawa Cancer Center, Yokohama 241-8515, Japan; s-okamoto@kcch.jp (S.O.); s-toda@kcch.jp (S.T.); ka0rik0hagura@gmail.com (K.K.); chandra_odj160@yahoo.co.jp (Y.M.); yu.x.shimada@gmail.com (Y.S.); mtintksh@msn.com (T.Y.); tyamashita@kcch.jp (T.Y.); 3Department of Nursing, Kanagawa Cancer Center, Yokohama 241-8515, Japan; 4Cancer Prevention and Control Division, Kanagawa Cancer Center Research Institute, Yokohama 241-8515, Japan; 5Graduate School of Health Innovation, Kanagawa University of Human Services, Kawasaki 210-0821, Japan

**Keywords:** HBOC, NCCN, screening, genetic testing, *BRCA1/2*

## Abstract

Clinical screening using the National Comprehensive Cancer Network (NCCN) testing criteria may fail to identify all patients with hereditary breast and ovarian cancers. Thus, this study aimed to evaluate the strategy of expanding target patients for genetic testing among Japanese patients. We reviewed the medical records of 91 breast cancer patients who underwent genetic testing. Among 91 patients, eight were diagnosed with pathogenic or likely pathogenic variants: *BRCA1* (n = 4) and *BRCA2* (n = 4). Among 50 patients meeting the testing criteria of the guidelines, 6 (12%) were diagnosed with pathogenic or likely pathogenic variants. The sensitivity and specificity of screening using the testing criteria were 75% and 47%, respectively. Expanding the NCCN criteria to include all women diagnosed with breast cancer aged ≤65 years achieved 88% sensitivity but 8% specificity. The expansion of the NCCN criteria could benefit Japanese patients; however, larger studies are necessary to change clinical practice.

## 1. Introduction

Breast cancer is one of the most common cancers in Japan, with approximately 90,000 new cases being diagnosed every year. Approximately 10,000 patients in Japan die from breast cancer each year [1]. Approximately 5–10% of breast cancers are strongly related to genetic background, of which hereditary breast and ovarian cancer (HBOC) are the most common. HBOC is diagnosed based on the presence of *BRCA1* or *BRCA2* (*BRCA1/2*) pathogenic germline variants. Carriers of *BRCA1* and *BRCA2* variants have cumulative breast cancer risks of 72% and 69%, respectively, and cumulative ovarian cancer risks of 44% and 17%, respectively, up to the age of 80 years, which is remarkably higher than those of the general population [2,3,4,5,6].

The National Comprehensive Cancer Network (NCCN) recommends the screening of patients at a high risk of HBOC, followed by germline genetic testing. Genetic testing is recommended for individuals with suspected HBOC and for the relatives of carriers of the *BRCA1/2* mutation. For example, an individual diagnosed with breast cancer at a young age or ovarian cancer at any age is recommended to undergo genetic testing. Similarly, a patient with breast cancer who has a family history of breast cancer or ovarian cancer is recommended to undergo genetic testing [7,8]. This practice is reasonable from a health economic perspective since genetic testing is rather expensive.

A study on HBOC conducted in the United States (US) showed that clinical screening using the NCCN criteria may fail to identify all patients with HBOC [9]. In particular, genetic testing was suggested for all patients with breast cancer aged <65 years, rather than screening with the testing criteria [9]. The cost of genetic testing is decreasing; correspondingly, the feasibility of this expansion strategy is also increasing. As shown in the aforementioned study in the US, it may also be possible to expand the target group for genetic testing in Japan. However, the prevalence of HBOC and the age at onset may differ between Japan and Western countries; thus, the usefulness of this strategy needs to be evaluated using data from Japan.

To evaluate the efficacy of expanding upon the NCCN criteria for clinical screening in Japan, prospective clinical studies on all patients with breast cancer, regardless of onset age or presence of family history, are needed. However, genetic testing is still expensive in Japan, indicating that such research would not be feasible. In Japan, *BRCA1/2* genetic testing has been available as a companion diagnostic testing for molecular target agents since 2018. Patients who are candidates for these drugs can undergo genetic testing without screening as required by the traditional testing criteria by NCCN, ref. [10], indicating that the expansion strategy has been adopted in daily practice. The number of patients who are eligible for genetic testing based on such criteria is gradually increasing. The efficacy of the expansion strategy for clinical screening based on the NCCN criteria can be retrospectively evaluated using this clinical data. Thus, we conducted this retrospective study to evaluate the conventional strategy for screening Japanese patients for genetic testing using the NCCN testing criteria, using clinical data of patients with breast cancer.

## 2. Patients and Methods

We retrospectively reviewed the medical records of 91 patients with breast cancer who underwent genetic testing for *BRCA1/2* as a companion diagnostic testing for olaparib (BRACAnalysis^®^ diagnostic system, provided by Myriad Genetics, Inc., Salt Lake City, UT, USA) at Kanagawa Cancer Center, from October 2018 to December 2019. We also reviewed the histories of the patients’ close blood relatives including their first- and second-degree relatives and their cousins. This companion diagnostic testing was approved for patients with HER-2-negative advanced or relapsed breast cancer by the health insurance in 2018 in Japan [11]. All 91 patients had HER-2-negative advanced or relapsed breast cancer, which was the indication for olaparib and the companion diagnostic system.

In clinical practice, we use the *BRCA1/2* testing criteria of the NCCN Guidelines^®^ for Genetic/Familial High-Risk Assessment: Breast and Ovarian Version 3.2019 after excluding patients who undergo the companion diagnostic testing for olaparib. This guideline demonstrates the criteria for further risk evaluation using genetic testing. We identify individuals at high risk of HBOC and recommend *BRCA1/2* genetic testing according to this guideline. In brief, it recommends that patients diagnosed with ovarian cancer, pancreatic cancer, or metastatic prostate cancer at any age undergo genetic testing. It also recommends genetic testing for those with (1) breast cancer diagnosed at ≤45 years of age, (2) triple-negative breast cancer diagnosed at ≤60 years of age, (3) two primary breast cancers diagnosed at ≤50 years of age, or (4) any family history of related cancers of HBOC. In our clinical practice, we use the *BRCA1/2* testing criteria that have been partially modified for our clinical practice at Kanagawa Cancer Center, as described elsewhere [12]. In the NCCN criteria, patients diagnosed with breast cancer aged <45 years are included in the high-risk group regardless of their family history. In this modification, the age was set to ≤40 years.

In this study, patients who underwent genetic testing for *BRCA1/2* as a companion diagnostic program for olaparib were retrospectively divided into two groups: those identified as high-risk patients for genetic testing using the *BRCA1/2* testing criteria in the NCCN Guidelines^®^ for Genetic/Familial High-Risk Assessment: Breast and Ovarian Version 3.2019 with our modification and those who did not meet these criteria [7,8] and were identified as the low-risk group. Patients with unknown family history or relatives with unknown cancer pathology who could not be determined to be in the high-risk group were included in the low-risk group.

Sensitivity and specificity were calculated to evaluate the usefulness of the guidelines. Similarly, the usefulness of the expansion of these guidelines was studied. The expanded criteria included meeting the NCCN testing criteria or being ≤50 years old, meeting the criteria or being ≤55 years old, meeting the criteria or being ≤60 years old, meeting the criteria or being ≤65 years old, meeting the criteria or being ≤70 years old, and meeting the criteria or being ≤75 years old. The sensitivity of the NCCN criteria in this study was calculated by dividing the number of patients with pathogenic or likely pathogenic variants who met the modified criteria by the number of all patients with pathogenic or likely pathogenic variants. The specificity was calculated by dividing the number of the patients without pathogenic or likely pathogenic variants who did not meet the modified criteria by the number of all the patients without pathogenic or likely pathogenic variants.

The relationship between meeting the criteria and testing results was evaluated using a Fisher’s exact test. Statistical analyses were performed using EZR on R commander, version 1.4(Y.Kanda, Jichi Medical University, Saitama, Japan) [13]. For all analyses, *p*-values < 0.05 were considered statistically significant. This study was approved by the institutional review board of the Kanagawa Cancer Center (2019-126).

## 3. Results

Among 91 patients with HER-2-negative advanced or relapsed breast cancer, including 2 men and 89 women, 8 were diagnosed with pathogenic or likely pathogenic variants: *BRCA1* (n = 4) and *BRCA2* (n = 4) (Table 1). All of the 8 patients were women. Among the 50 patients who met the testing criteria of the guidelines, 6 (12%) were diagnosed with pathogenic or likely pathogenic variants. Among 41 patients who did not meet the testing criteria, 2 (4.9%) were diagnosed with pathogenic or likely pathogenic variants. Details of breast cancer subtype and personal history of cancers are shown in Table 1.

No statistically significant relationship was found between meeting the criteria of the NCCN Guidelines^®^ and the genetic test results of BRACAnalysis^®^ diagnostic system (odds ratio 2.6; *p* = 0.28). Two patients of the eight with pathogenic or likely pathogenic variants did not meet the criteria. One patient with a *BRCA1* pathogenic variant was diagnosed with triple-negative, unilateral breast cancer when she was 72 years old. She has a sister who developed breast cancer in her early 60s. The other patient was diagnosed with hormone receptor-positive breast cancer when she was 62 years old. Her family history did not include breast cancer, ovarian cancer, pancreatic cancer, or prostate cancer.

Table 2 shows the sensitivity and specificity of *BRCA1/2* testing criteria of the NCCN Guidelines^®^ for Genetic/Familial High-Risk Assessment: Breast and Ovarian Version 3.2019 with our modification, which was calculated using details of the 91 patients. The sensitivity and specificity of screening using the testing criteria were 75% and 47%, respectively. Expansion of the conventional NCCN criteria to include all patients diagnosed with breast cancer and aged ≤65 years achieved 88% sensitivity; however, the specificity was 8% (Table 2). It could identify 7 of the 8 patients with pathogenic or likely pathogenic variants as high-risk patients for genetic testing. The expansion of the criteria to include all patients diagnosed with breast cancer and aged ≤75 years could identify all the patients with pathogenic or likely pathogenic variants as high-risk patients (Table 2).

## 4. Discussion

This study showed that some patients with *BRCA1/2* pathogenic variants or likely pathogenic variants could be missed by screening using the NCCN testing criteria in Japan, consistent with a previous study in the US [9]. Notably, two out of eight patients with HBOC could be missed according to the conventional NCCN criteria. The expansion of NCCN criteria is worth investigating in Japanese patients, including the provision of genetic testing for all patients under a certain age. One optimal age threshold is 65 years, with more favorable sensitivity than that of the conventional NCCN testing criteria. However, specificity was more unfavorable in this strategy, indicating that more patients without *BRCA1/2* pathogenic variants would receive genetic testing. Nonetheless, this could be acceptable in clinical practice because genetic testing is not physically invasive for patients, although future large-scaled studies to confirm the findings of this study are required.

Furthermore, the optimal age threshold for providing genetic testing for all patients warrant further discussion. In this study, a higher threshold for age indicated lower specificity; the conventional criteria would only miss two of the eight patients with HBOC. The expansion of NCCN testing criteria plus ≤65 years old would miss one patient, and the expansion of NCCN testing criteria plus ≤75 years old would miss no patients. An investigation from a health economic perspective is required to evaluate the usefulness of this expansion strategy. Among those who met the NCCN testing criteria and those who did not, testing about 9 (8.3) and 21 patients, respectively, was necessary to identify one patient with *BRCA1/2* pathogenic or likely pathogenic variant. With health insurance in Japan, genetic testing for *BRCA1/2* costs JPY 202,000 (USD 1867). This indicates that it costs JPY 1,676,600 (USD 15,500) and JPY 4,242,000 (USD 39,207) to identify one patient with *BRCA1/2* pathogenic or likely pathogenic variants, respectively. The lower threshold of age indicates a lower cost of identifying one patient. Therefore, when deciding on the age range, we must consider the health economic and public health aspects of this expenditure.

This study has important implications in future clinical practice; however, some limitations exist. First, this study was conducted in a single center and the sample size was small. Future large-scaled studies to confirm the findings of this study are needed. This would help deduce the optimal age threshold for genetic testing for all patients. Secondly, this study included patients with advanced or relapsed breast cancer. Usually, many patients with breast cancer undergo genetic testing before surgery. Thus, the backgrounds of patients in this study and those undergoing genetic testing in routine medical practice may differ; the ratio of triple-negative breast cancer could similarly be higher in this study [14]. Finally, the cost of genetic testing is expected to drastically decrease in the future. Additional cost analyses using these decreased costs will be required.

## 5. Conclusions

The expansion of NCCN criteria could benefit Japanese patients by additionally providing genetic testing for all patients under a certain age, preferably 65 years; however, larger studies are necessary to change clinical practice.

## Figures and Tables

**Table 1 ijerph-19-06182-t001:** Characteristic of the patients.

Characteristic	Total (N = 91)	Meeting NCCN Criteria * (n = 50)	Not Meeting NCCN Criteria (n = 41)
Age at diagnosis of first breast cancer, years			
Median	52 ***	48 ***	58
≤29	2	2	0
30–39	14	14	0
40–49	26	15	11
50–59	24	10	14
60–69	17	7	10
70–79	8	2	6
Subtype **			
Triple-negative	23	13	10
Luminal	68	37	31
Personal history of other cancers			
Any cancer	5	2	3
Ovarian	0	0	0
Pancreatic	1	1	0
*BRCA1/2*			
*BRCA1* pathogenic or likely pathogenic variants	4	3	1
*BRCA2* pathogenic or likely pathogenic variants	4	3	1
No pathogenic or likely pathogenic variants	83	44	39

* We used revised criteria. Patients with unknown family history or relatives with unknown cancer pathology who could not be determined to be in the high-risk group were included in the low-risk group. According to the NCCN criteria, patients with a diagnosis of breast cancer aged <45 years were included in the high-risk group regardless of family history. In this modification, this age was set to ≤40 years. ** All of the 91 patients had HER-2 negative breast cancer. *** One patient was excluded due to lack of exact information regarding age.

**Table 2 ijerph-19-06182-t002:** Sensitivity and specificity.

	Patients with Pathogenic or Likely Pathogenic Variants	Patients without Pathogenic or Likely Pathogenic Variants	Sensitivity	Specificity
Meeting NCCN testing criteria * or age at diagnosis, years				
Meeting NCCN testing criteria	6	44	0.75	0.47
Not meeting NCCN testing criteria	2	39		
Meeting NCCN testing criteria or ≤50 years old	6	55	0.75	0.34
Not meeting NCCN testing criteria and >50 years old	2	28		
Meeting NCCN testing criteria or ≤55 years old	6	61	0.75	0.27
Not meeting NCCN testing criteria and >55 years old	2	22		
Meeting NCCN testing criteria or ≤60 years old	6	71	0.75	0.14
Not meeting NCCN testing criteria and >60 years old	2	12		
Meeting NCCN testing criteria or ≤65 years old	7	76	0.88	0.08
Not meeting NCCN testing criteria and >65 years old	1	7		
Meeting NCCN testing criteria or ≤70 years old	7	78	0.88	0.06
Not meeting NCCN testing criteria and >70 years old	1	5		
Meeting NCCN testing criteria or ≤75 years old	8	81	1.00	0.02
Not meeting NCCN testing criteria and >75 years old	0	2		

* We used the modified criteria.

## Data Availability

The date used in this study cannot be made publicly available. We did not obtain informed consent to make the date publicly available from the patients.

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
