# Peer review of "Efficacy of Clinical Guidelines in Identifying All Japanese Patients with Hereditary Breast and Ovarian Cancer"

_ijerph, 2022, doi:10.3390/ijerph19106182_

Round 1
Reviewer 1 Report
The topic of the article is with significant implications for medical practice. Screening for breast and ovarian cancer is essential for early diagnosis and treatment, especially in people with a genetic background.
We recommend that you complete the Section Discussions with several aspects according to the recommendations in the Instructions for Authors: Authors should discuss the results and how they can be interpreted from the perspective of previous studies and working hypotheses. The findings and their implications should be discussed in the broadest possible context, and the limitations of the work should be highlighted. Future research directions may also be mentioned. This section may be combined with Results.
The article presents the results of a single medical centre and other limitations, but it illustrates the authors' research results.
Reviewer 2 Report
This is a small, single institution study that could serve as background justification for larger more definitive study. I suggest adding this the abstract last sentence-"but larger studies are necessary to change clinical practice". The discussion is appropriately cautious in applicability of the small data set. One small comment-lines 56-57 have a comma splice.
n
Reviewer 3 Report
" Efficacy of clinical guidelines in identifying all Japanese pa-2 tients with hereditary breast and ovarian cancer" is a nice work for Japanese. It is a well-structured and well-written paper that could be considered for publication. I have only one suggestion. Would the author suggest how to strategy for clinical or public health of the low specificity (8%). Please discuss in discussion section.
Round 2
Reviewer 2 Report
The authors have answered my concerns.The paper is much improved.
This manuscript is a resubmission of an earlier submission. The following is a list of the peer review reports and author responses from that submission.